# The Handbook of Minerals on a Gluten-Free Diet

**DOI:** 10.3390/nu10111683

**Published:** 2018-11-05

**Authors:** Iga Rybicka

**Affiliations:** Faculty of Commodity Science, Poznań University of Economics and Business, 61-875 Poznań, Poland; iga.rybicka@ue.poznan.pl

**Keywords:** gluten-free diet, mineral, deficiency, calcium, iron, magnesium, zinc

## Abstract

The importance of a gluten-free diet (GFD) in the treatment of celiac disease and other gluten-related disorders is undisputable. However, strict GFD often lead to nutritional imbalances and, therefore, to deficiencies. One of the most common deficiencies from a GFD are an insufficient amount of Ca, Fe, Mg, and Zn. This is mainly because the most of popular gluten-free (GF) raw materials are poor in minerals. Although the popularity of GFD is constantly growing, the data on minerals in GF products are still limited. More importantly, an access to the data is even more restricted. Therefore, the paper reviews the Ca, Fe, Mg, and Zn contents in hundreds of grain GF products available worldwide. The data for 444 products from categories of flours, mixes for cooking, bakery products, cereals, groats, rice, and pasta are obtained from research papers and nutritional databases. The calculation of the realization of mineral requirements from a portion of each product with its graphical classification as rich/average/poor source of each mineral is given. The review is a handbook of minerals for people on a GFD, dietitians, and food producers.

## 1. Introduction

A gluten-free diet (GFD) has been the most popular elimination diet for more than a decade. The number of people on a GFD is constantly increasing [1]. It results primarily from a “free from” trend [2] and not from the higher prevalence of gluten-related disorders [3]. In the United States, more than 100 million people consume gluten-free (GF) products with most of these people lacking any gluten-related disorders [4].

Although GFD is associated with being more healthy [5], epidemiological studies indicate nutritional imbalances for people following GFD. They refer both to macronutrients and micronutrients including minerals. Vici [6] and Gobbetti [7] with coauthors reviewed data from clinical studies on a nutritional status on a GFD indicating too high intake of energy, sugars, lipids, and saturated lipids and too low intake of fiber, vitamin D, B-group vitamins, and minerals—Ca, Fe, Mg, and Zn.

Despite the huge popularity of GFD, the data on mineral content in GF products are still limited. More importantly, an access to such data is even more restricted. The improvement of the nutritional quality of GFD, which is mentioned by many researchers [7,8,9,10], should start in the quality assessment of available products. The constant expansion of the GF nutritional data together with alimentary education are key elements for successful therapy in gluten-related disorders. Therefore, the aim of the study was to develop a database of Ca, Fe, Mg, and Zn in grain gluten-free products available worldwide. The results are presented in separate tables for each mineral in a way that allows quick recognition for the products of high content of the nutrient from different categories. The review is a handbook of minerals not only for people on a gluten-free diet but also for food producers and dietitians who play a crucial role in the education of patients with gluten-related disorders.

## 2. Materials and Methods

### 2.1. Methods

The data were electronically searched for keywords in different national databases and literature data. The keywords included in the study were: “gluten-free” (“gluten free”) plus “mineral” (including “calcium” or “iron” or “magnesium” “or “zinc” or “element” or “microelement” or “macroelement”). The study was restricted to the publications after 1 January 2005.

Data from nutritional databases are presented as they were found. When producer or ingredients were given, they are presented. The data obtained from research articles included only products with detailed material’s description or ingredients or the recipe.

### 2.2. Data Extraction

Data selection under the study is presented in Scheme 1.

Literature references were searched in the Scopus while the nutritional databases under the study were those found on the FAO website [11]. The study includes databases from the United States [12,13], Australia [14], Canada [15], and several European countries [16,17,18,19,20,21]. Records with at least one mineral of interest were analyzed. All commercial products selected from the nutritional databases were described as “gluten-free” or were signed with the Cross Grain symbol. Products from research studies had to be described as GF. Raw materials and naturally gluten-free products not described as a GF, e.g., rice, buckwheat, and groats were omitted.

## 3. Results & Discussion

A total of 444 GF products were incorporated into the study including 186 products from literature data (23 articles and book chapters) and 258 GF products from 10 databases. Databases included in the study were from the United States (212 products) [12,13], Germany (14 products) [16], Norway (10 products) [17], Holland (7 products) [18], Australia (6 products) [14], Finland (6 products) [19], Belgium (2 products) [20], Canada (2 products) [15], and Estonia (1 product) [21].

No GF hits for minerals of interest were found in databases form: Armenia [22], the Czech Republic [23], Denmark [24], France [25], Greece [26], Iceland [27], Italy [28], Latvia [29], Serbia [30], Slovakia [31], Spain [32], Sweden [33], Switzerland [34], Turkey [35], and the United Kingdom [36].

The handbook replies only to grain GF products since they should partially consist of the everyday diet. Even though grain products are not the best source of all analyzed minerals (e.g., Ca), due to their substantial daily consumption, they significantly realize daily requirements for nutrients.

Moreover, different studies indicate that the proportion of the main nutrients on GFD is improper and that people on a GFD often omit some assortment of grain products like commercial bakery products [37]. Only grain products that should consist of the everyday diet were included in the study. Cookies and snacks, e.g., crackers, were not taken into account. The total number of grain GF products was significant. Therefore, they were divided into categories shown below.
I—floursII—mixes for cookingIII—bakery products (e.g., breads, rolls, dinner rolls, crack bread)IV—cereals (plain and musli flakes)V—grains and riceVI—pasta

All included products were described as GF. When the manufacturer or a brand was known, the ingredients were not presented since consumers can easily recognize these products. When the producer was unknown, the main ingredients were presented (if applicable).

The daily portion for each category was established. For the I, II, and IV category, it was 30 g, for V and VI, it was 50 g, and, for III, it was 100 g. The percent of Reference Nutrient Intakes (RNIs) were calculated for 800 mg, 14 mg, 375 mg, and 10 mg for Ca, Fe, Mg, and Zn, respectively [38].

Moreover, different colors were given to show the contribution of the portion of each product to realization of daily requirements for minerals (Figure 1). Dark green indicated that these products are a good source of nutrients (>25% of RNI), bright green indicates a moderate source (10% to 25% of RNI), and white indicates a poor source (<10% of RNI).

Appendix A present the data for Ca, Fe, Mg, and Zn, respectively, in a range of GF products separately for each category. The data were expressed in mg for 100 g of the edible product (fresh matter). The data were sorted from the highest to the lowest content of each mineral. If original data are presented in dry matter and the moisture is given, data were calculated into fresh matter. The authors of dry matter data were asked for water content data. When they did not answer or the data were unavailable, the data were calculated for the average moisture of 10% for VI category, 15% for I, II, IV, and V category, and 40% for III category [39,40]. The data for equal products from different references were averaged, e.g., the content of Ca in amaranth flour presented in Appendix A was calculated from values from four references. For these products, the minimum and maximum contents were presented.

As presented in Appendix A, only three products were characterized by high content of Ca. These were crisp bread [17] and white and brown breads included in the Dutch database [18]. Additionally, 34 were of moderate contribution to the daily intake of Ca and most of them were bakery products. No data for calcium were found for a quarter of GF products (113 products). As mentioned above, because of significant daily consumption of grain products, they contribute to the daily intake of Ca even though they are not considered a good source of that mineral.

Databases and literature references were well-supplied with Fe data. Only 36 out of 444 products did not deliver data on iron content (Appendix A). A total of 23 products were classified as an important source of Fe including most from categories of bakery products and pasta. Additionally, 112 products from all categories were of moderate contribution to daily intake of Fe.

In addition, 63% of selected products (278 items) were not characterized by the content of Mg (Appendix A). From the remaining 166 products, only six bakery products (e.g., crisp breads) were classified as an important source of magnesium [12,14,16,18,19]. A total of 53 products for the I, III, IV, and VI category were of moderate content of Mg. None of the products from the II and V category were regarded as of high or moderate impact on the realization of Mg requirements.

Most of the products (372 items) did not deliver data on Zn content (Appendix A). Only eight products (mostly crisp breads) were classified as of high impact on daily intake of zinc [12,17,19,41,42,43]. Twenty-nine products from 5 out 6 categories were described as a moderate source of Zn. All products from category II were assigned as a poor source of zinc.

The content of minerals in food product primarily results from its ingredients especially raw materials. Thus, in Table 1, contents of Ca, Fe, Mg, and Zn in the most popular GF raw materials—amaranth, buckwheat, chickpea, corn, millet, quinoa, rice, sorghum, tapioca, and teff were given. Original data expressed in dry matter were calculated into fresh matter at the level of 15% water [44]. As shown in Appendix A, most of the GF products available in the worldwide market were produced from rice and corn, which are not a good source of macro-elements and microelements. Pseudo cereals like amaranth, quinoa, or teff are e.g., excellent sources of Fe and Mg, but their characteristic aroma and flavor limit their application in significant quantities in food products. For example, the most preferable amount of teff in breads should not exceed 10% [45]. Many researchers underline the necessity of the improvement of nutritional quality of GFD by replacing the low-nutritional GF raw materials with pseudo cereals with high nutritive value or by fortification with minerals [7,8,9,10]. The presented data on the mineral composition in commercial GF products (Appendix A) along with information on the abundance of cereals and pseudo cereals in minerals (Table 1) prompt how to mix them in order to obtain nutritionally balanced food or meal. For example, 5% of an addition of acorn flour to the Gluten Free Cornbread Mix produced by The King Arthur Flour Company Inc. (Norwich, VT, USA) [12] will enrich it in iron by approximately 10%.

Nevertheless, review on recent literature revealed several important drawbacks. One of the most worrisome is the lack of data on Mg and Zn contents in the majority of GF products. Most of the total 444 foods included in the study did not have data on zinc (84% products). The data on magnesium was also very limited. In fact, 63% of products had no data on that mineral. Another problem was the quality of some of the presented results. Data presented from the largest database, USDA, is open for everybody and it allows manufacturers to describe their products. It can lead to the overestimation of their products. Thus, the ideal situation is building the databases based on the analytical determinations like those presented by e.g., Mazzeo and co-authors [56].

In addition, the problem is that the average consumer does not have access to literature references. Therefore, the actualization of online nutritional databases is the most preferable way to achieve this goal. The data should be developed at national levels and should be easily accessed. Nutritional databases should be free, on-line, and regularly updated. Providing the free-of-charge nutritional databases should be one of the key responsibilities for national food and nutrition institutes. The availability and quality of nutritional databases is even more crucial for the assortment of GF products since GFD is the only efficient treatment for celiac disease and other gluten-related disorders.

## 4. Conclusions

The handbook of minerals on GFD allows for the quick recognition of the best sources of Ca, Fe, Mg, and Zn. These are elements that are often deficient when excluding gluten. When analyzing the mineral composition of GF grain products, it is common to identify that they are rarely rich or moderate source of minerals. It applies in particular to the most popular GF products made with starches and refined flours. What is even more disturbing is the limited data on Mg and Zn contents in GF products. Additionally, access to this information is very restricted. The easily accessed nutritional databases for GF products should by developed at a national level and at international levels. Only then nutritional education, which is a crucial element of treating gluten-related disorders, would be credible and effective.

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
