# Peer review of "The Handbook of Minerals on a Gluten-Free Diet"

_nutrients, 2018, doi:10.3390/nu10111683_

Round 1
Reviewer 1 Report
The paper is nicely written and addresses a key issue in gluten-free nutrition councelling
Author Response
Dear Reviewer,
thank you for your opinion.
Yours faithfully,
Iga Rybicka
Reviewer 2 Report
-Very concise and interesting. Raises the point about lack of info about nutritional deficiencies in GF products.
-Last sentence of the ABSTRACT means exactly the opposite of what they want to say ("...not only for people on a GFD but also for...")
-Line 83: "brand was unknown"?
-Visualize recommended DV for these minerals vs. contribution of GF products
-Discussion: what about supplementing GF products with these minerals?
Author Response
Response to Reviewer 2
Dear Reviewer,
thank you for comments and suggestions. The manuscript has been corrected according to your indications and the point-by-point response to your comments is presented below.
Yours faithfully,
Iga Rybicka
Point 1: Very concise and interesting. Raises the point about lack of info about nutritional deficiencies in GF products.
Response 1: Thank you.
Point 2: Last sentence of the ABSTRACT means exactly the opposite of what they want to say ("...not only for people on a GFD but also for...").
Response 2: The sentence was modified into: “The review is a handbook of minerals for people on a GFD, dietitians, and food producers.”
Point 3: Line 83: "brand was unknown"?
Response 3: The paragraph was modified from “All included products were described as GF. When manufacturer or a brand was known, the ingredients were not presented.” into “All included products were described as GF. When manufacturer or a brand was known, the ingredients were not presented, as these products can be easily recognized by consumers. When producer was unknown the main ingredients were presented (if applicable).”
Point 4: Visualize recommended DV for these minerals vs. contribution of GF products Response 4: Reference Nutrient Intakes (RNIs) are indicated in the text, while contribution of GF products is presented in Supplementary Materials. I think that because of the large amount of data (dozens of pages included in the Supplementary Materials) tables are the most transparent form of results presentation.
Point 5: Discussion: what about supplementing GF products with these minerals?
Response 5: I agree that fortification could be a solution, so I modified sentence: “Many researchers underline the necessity of improvement of nutritional quality of GFD by replacing the low-nutritional GF raw materials with pseudocereals with high nutritive value [7-10].” into “Many researchers underline the necessity of improvement of nutritional quality of GFD by replacing the low-nutritional GF raw materials with pseudocereals with high nutritive value or by fortification it with minerals [7–10].”

Reviewer 3 Report
The manuscript by Rybicka, I. entitled „The handbook of minerals on a gluten-free diet“ is a compilation of the minerals Ca, Fe, Mg, and Zn, of a variety of different gluten-free products. Since these products are free of wheat, rye or barley, which are all cereals with naturally high content of minerals, the gluten-free diet is rather poor in minerals. This summary is highly welcome to point out products with low mineral concentration and to raise the awareness of trained staff for alternative or more suitable products.
There are some minor remarks:
Abstract, line 16: Please amend the last sentence and cancel “not”
Line 56: Literature was also searched in databases from Germany, Norway, Holland….
Line 79: Please explain why cookies were not included in bakery products.
Line 83: Please add a short explanation why ingredients were not presented when manufacturer was known.
Line 124: Ref 64 should be corrected into Ref 63
Table 1: To make the table clearer, lines should be integrated between subdivisions.
S1-S4: In daily life the dietician has no references available. It would therefore be more appropriate to use averages of similar products especially when no manufacturer is named, e.g. Amaranth Flour (group I) is named four times with Ref 39,40,41,42 or Granola Garden of Light Inc (group IV) is named three times.
Author Response
Response to the reviewer’s comments is attached as a PDF file.
